# Malnutrition Prevalence according to the GLIM Criteria in Head and Neck Cancer Patients Undergoing Cancer Treatment

**DOI:** 10.3390/nu12113493

**Published:** 2020-11-13

**Authors:** Belinda Steer, Jenelle Loeliger, Lara Edbrooke, Irene Deftereos, Erin Laing, Nicole Kiss

**Affiliations:** 1Nutrition and Speech Pathology Department, Peter MacCallum Cancer Centre, Melbourne, VIC 3000, Australia; jenelle.loeliger@petermac.org (J.L.); Erin.laing@petermac.org (E.L.); 2Allied Health Department, Peter MacCallum Cancer Centre, Melbourne, VIC 3000, Australia; lara.edbrooke@petermac.org (L.E.); nicole.kiss@deakin.edu.au (N.K.); 3Physiotherapy Department, The University of Melbourne, Parkville, VIC 3052, Australia; 4Department of Surgery, Western Health, The University of Melbourne, Melbourne, VIC 3000, Australia; irene.deftereos@unimelb.edu.au; 5Department of Nutrition and Dietetics, Western Health, Footscray, VIC 3011, Australia; 6Institute for Physical Activity and Nutrition, Deakin University, Geelong, VIC 3220, Australia

**Keywords:** head and neck neoplasms, malnutrition, nutrition

## Abstract

Malnutrition is highly prevalent in people with head and neck cancer (HCN) and is associated with poorer outcomes. However, variation in malnutrition diagnostic criteria has made translation of the most effective interventions into practice challenging. This study aimed to determine the prevalence of malnutrition in a HNC population according to the Global Leadership Initiative on Malnutrition (GLIM) criteria and assess inter-rater reliability and predictive validity. A secondary analysis of data available for 188 patients with HNC extracted from two cancer malnutrition point prevalence studies was conducted. A GLIM diagnosis of malnutrition was assigned when one phenotypic and one etiologic criterion were present. Phenotypic criteria were ≥5% unintentional loss of body weight, body mass index (BMI), and subjective evidence of muscle loss. Etiologic criteria were reduced food intake, and presence of metastatic disease as a proxy for inflammation. The prevalence of malnutrition was 22.6% (8.0% moderately malnourished; 13.3% severely malnourished). Inter-rater reliability was classified as excellent for the GLIM criteria overall, as well as for each individual criterion. A GLIM diagnosis of malnutrition was found to be significantly associated with BMI but was not predictive of 30 day hospital readmission. Further large, prospective cohort studies are required in this patient population to further validate the GLIM criteria.

## 1. Introduction

Head and neck cancer (HNC) had an age-standardised incidence rate of 16.6 cases per 100,000 persons in Australia in 2016 [1]. It is well established that the prevalence of malnutrition in people with HNC is one of the highest of all cancer types, ranging from 20 to 74% when assessed by various methods [2,3,4,5,6,7,8]. A tumour in the head and neck region can result in significant symptoms affecting food intake, including dysphagia, trismus and odynophagia [9], which, when combined with the toxicities caused by cancer treatment [10], contributes to the high rate of malnutrition. Malnutrition in HNC, when assessed by various methods, leads to increased complications [11,12,13], hospital length of stay [11,12,13] and readmission rates [12], and reduced treatment response [13], quality of life [14] and survival [11,13].

The variation in malnutrition prevalence in people with HNC may be explained by the heterogeneity of the cohorts studied, including stage of disease, at varying stage of diagnosis, and different treatment modalities. The variation may also be explained by the numerous criteria used to diagnose malnutrition. Percentage loss of weight [3,5,15], body mass index (BMI) [3,5,15], serum albumin levels and skin fold thickness [3], either on their own or in combination, were common criterion used in earlier studies reporting the prevalence of malnutrition in people with HNC. More recently, studies have used validated nutrition assessment tools such as the Patient-Generated Subjective Global Assessment (PG-SGA) [4,6,7,16,17]. The variation in criterion makes comparison of the effectiveness of nutrition interventions across different studies challenging. A clinically relevant global malnutrition definition is required to ensure robust and comparable reporting of intervention outcomes to support translation of the most effective interventions into practice.

In 2018, the Global Leadership Initiative on Malnutrition (GLIM) published a global consensus-based definition of malnutrition [18] that is suitable for all adults, healthcare settings and medical specialties. The GLIM criteria to define malnutrition involves two steps: (1) screening for malnutrition risk using a validated screening tool to determine at-risk status, followed by (2) assessment for diagnosis and severity grading [18]. A diagnosis of malnutrition requires at least one phenotypic criterion, including non-volitional weight loss, low BMI, or reduced muscle mass, and one etiologic criterion including reduced food intake or disease burden/inflammation to be present [18]. The severity grading is based on the degree of loss for the phenotypic criteria [18]. Following the development of the GLIM criteria, further studies are required to establish its psychometric properties, including validity and reliability across different clinical populations [19].

Currently, only one study has used the GLIM criteria to diagnose malnutrition in HNC patients [8]. This study by Einarsson et al. prospectively determined the prevalence of malnutrition at the beginning, during and post-treatment in a heterogenous cohort of HNC patients undergoing a variety of treatments. Objective measures were used (including fat free mass index and C-reactive protein) to determine a diagnosis of malnutrition using the GLIM criteria, with results demonstrating a malnutrition prevalence of between 0.5 and 32.4% depending on the time point and criterion combination used. The GLIM criteria have been developed to encompass a variety of measures for each criterion that are used across different practice settings, including both objective and subjective measures of muscle stores [18]. This was considered important as objective measures are not always available or practical in the clinical setting. A commonly utilised tool for subjective assessment of muscle stores in acute oncology practice is the PG-SGA, and therefore validation of the GLIM criteria in HNC using this tool is required. Additionally, no studies have investigated the ability of the GLIM criteria to predict clinically relevant outcomes in HNC, or inter-rater reliability; both of which are required to further investigate the applicability of GLIM criteria in clinical practice [19]. Therefore, this study aimed to determine (1) the prevalence of malnutrition in HNC patients according to the GLIM criteria, using PG-SGA as a subjective assessment of muscle stores; (2) the inter-rater reliability of using the GLIM criteria to identify malnutrition; and (3) associations between the GLIM criteria and demographic and clinical characteristics and the predictive validity of the GLIM criteria with respect to hospital readmission.

## 2. Materials and Methods

### 2.1. Study Design and Setting

This study was a secondary analysis using previously collected data from two multi-centre cancer malnutrition point prevalence studies conducted in 2016 and 2018. These studies were undertaken in participating public and private Victorian health services in Australia: 16 sites in 2016 and 19 sites in 2018.

Data collection was conducted over a common four-week period for each study: November to December 2016 and July to August 2018. At each participating health service, experienced clinical dietitians or trained dietitian students who had passed their clinical competencies, collected data for multi-day inpatients, ambulatory radiotherapy patients and chemotherapy day patients at both time points.

### 2.2. Participants

Eligible participants were aged 18 years and over; admitted to hospital for cancer treatment (including surgery) or related management for a minimum of 2 nights; or ambulatory patients attending for radiotherapy or intravenous chemotherapy or immunotherapy. 

Patients were excluded when they were attending for oral chemotherapy or maintenance/hormonal treatment only; were attending for day surgery or medical review only; were admitted to an ICU or emergency department on the day of data collection; were terminally ill with a life expectancy of less than one month; or were unable to provide verbal consent.

Only HNC patients recruited to the point prevalence study were included in this secondary analysis.

### 2.3. Variables

#### 2.3.1. Demographic and Clinical Data

Demographic data collected for each participant included age, sex and living situation (alone or with others). Clinical data included presence of metastatic disease, current cancer treatment (either alone or any combination of surgery, chemotherapy, radiotherapy, immunotherapy, other, or cancer-related management), treatment setting (inpatient or ambulatory), and treatment location (metropolitan or regional).

#### 2.3.2. Malnutrition Screening

Each participant was screened for malnutrition risk using the malnutrition screening tool (MST) [20,21], validated for use in the oncology population [22,23]. An MST score of 2 or greater indicated that the participant was at risk of malnutrition. Participants with an MST score less than 2 were assumed to be well nourished.

#### 2.3.3. Anthropometry

Body mass index (BMI) was calculated from height and weight recorded using usual equipment within each hospital or as reported by the participant. BMI was categorised as underweight (<18.5 kg/m^2^ if <65 years or <22 kg/m^2^ if ≥65 years), healthy weight (18.5–24.9 kg/m^2^ if <65 years or 22–27 kg/m^2^ if ≥65 years), and overweight (≥25 kg/m^2^ if <65 years or >27 kg/m^2^ if ≥65 years) [24,25]. If a participant reported weight loss when completing the MST, the amount of weight loss, time frame (≤3 months and ≥4 months) and whether the weight loss was unintentional were recorded.

Participants identified as being at risk of malnutrition according to the MST (score ≥2) had a subjective assessment of muscle stores completed on a minimum of 4 of the 7 muscle sites specified in the PG-SGA tool [26]. A minimum of 4 muscle sites were specified, acknowledging positioning may prevent assessment of all 7 sites. Muscle stores were graded as no deficit, mild/moderate deficit, or severe deficit for each assessed site.

#### 2.3.4. Food Intake Assessment

Participants were asked whether their food intake had reduced. Where a reduction in food intake was reported, the degree of reduction compared to usual intake (>75%, ≤75%, ≤50% and ≤25%) and duration (0–4 days, 5–30 days and >1 month) was recorded.

#### 2.3.5. Inflammation

As no objective marker of inflammation was available, the presence of metastatic disease was collected from the medical record and used as a proxy for inflammation. The blanket assignment of inflammation due to a cancer diagnosis is not recommended to be used for this etiologic criterion as it does not indicate the severity of the disease burden [19].

#### 2.3.6. GLIM Criteria

The GLIM criteria were applied by two independent dietitians, both with over 10 years clinical experience in oncology nutrition, using the three phenotypic and two etiologic criteria. How the data from the present study were applied using the GLIM criteria is presented in Table 1. In this study, malnutrition was diagnosed using two different combinations of GLIM criteria. Firstly, a diagnosis of malnutrition was confirmed when at least one phenotypic and one etiologic criterion were present (method 1). Secondly, recognising the limitations of the use of a proxy measure, a diagnosis of malnutrition excluding the presence of metastatic disease (proxy inflammation criterion) was also determined (method 2).

Severity of malnutrition, moderate or severe, was graded according to the extent of weight loss, BMI cut offs and extent of muscle deficit as described in Table 2.

#### 2.3.7. Outcomes

Thirty days following the initial data collection, data were collected from the medical record on any unplanned admissions to the same hospital and patient status (alive or deceased) for each participant.

#### 2.3.8. Statistical Analysis

Data analysis was completed using SPSS Statistics version 26. Descriptive statistics for all continuous variables were reported as mean (standard deviation) or median (inter-quartile range), depending on the normality of distribution. Categorical variables were reported as counts and percentages. Subsequent to descriptive reporting, all analyses used the GLIM variable according to the first method of assessment, whereby malnutrition was diagnosed by the presence of at least one phenotypic and one etiologic criterion being present. Construct (discriminant) validity was assessed using Chi-square tests for categorical variables and independent t-tests or Mann-Whitney U tests for continuous variables, dependent on the normality of distributions. A p-value of *p* < 0.05 indicated statistical significance for all tests. Inter-rater reliability was assessed using the Kappa statistic. Values greater than 80% represent ‘excellent’ agreement, >60% ‘substantial’, 40–60% ‘moderate’ and <40% ‘poor to fair’ agreement [27].

Binomial logistic regression models were used to determine the predictive validity of the GLIM criteria malnutrition diagnosis for unplanned hospital admissions up to 30 days post initial data collection. Potential co-variates to be considered in the models included: sex, age (continuous variable), social support (lives alone versus lives with carer/family), treatment setting (inpatient versus ambulatory), presence of metastatic disease and treatment location (metropolitan versus regional). Where variables were highly intercorrelated (multi-collinearity) one variable was removed from the model. Singularity was also assessed between the metastatic disease and malnutrition variables given metastatic disease is a criterion which forms part of the GLIM score. The malnutrition variable was retained in all models given it was the independent variable of interest. Findings are presented as odds ratios (95% confidence intervals). The recommended sample size for regression analyses is 50 + 8m (where ‘m’ represents the number of independent variables). Therefore, the outlined analyses required a complete case sample of 106 patients [28]. 

#### 2.3.9. Ethics Approval

Ethics approval was obtained from the Human Research Ethics Committee at Peter MacCallum Cancer Centre, Melbourne, Australia (HREC/16/PMCC/149.) with site specific approval obtained at each of the participating sites. This study was reported according to the Strengthening the Reporting of Observational Studies in Epidemiology (STROBE) guidelines [29].

## 3. Results

### 3.1. Participants

There were a total of 188 HNC participants in the combined 2016 and 2018 datasets. Of these, 177 participants had the required data to be able to determine the presence of malnutrition according to the first method of scoring and 184 when the metastatic disease (inflammation proxy) criterion was removed (see flowchart, Figure 1).

Participant characteristics for the complete dataset (*n* = 188) are reported in Table 3. The mean (standard deviation) age was 63.8 (12.4) years, the majority of participants were male (73.9%) and lived with family, a carer or in residential care (79.8%). Metastatic disease was present in 13.3% of participants. More than three-quarters of participants (77.7%) were receiving radiotherapy and almost half chemotherapy (45.7%) or surgery (42.6%). More than three-quarters of the participants (78.7%) were being treated in the ambulatory setting.

### 3.2. Prevalence of Malnutrition Risk and Malnutrition

Table 4 shows the prevalence of malnutrition risk according to the MST, and malnutrition according to the GLIM criteria (method 1) described in Table 1. Forty-two percent of participants were classified as at risk of malnutrition. The overall prevalence of malnutrition according to the GLIM criteria (method 1) was 22.6%, with 8.0% diagnosed as having moderate malnutrition and 13.3% having severe malnutrition.

The overall malnutrition prevalence according to the GLIM criteria, omitting the use of metastatic disease as a proxy criterion for inflammation (method 2) and using only the presence of reduced food intake as the etiologic criterion, was 20.7% (38/184). There were also an additional 18 participants who had at least one phenotypic criterion present but were not diagnosed as malnourished as neither etiologic criterion were present. When assessing subgroups by treatment modality, the prevalence of malnutrition using GLIM criteria was 24.1% (19/79) for surgery, 19.8% (17/86) for chemotherapy, 17.9% (26/145) for radiotherapy, and 0% (0/3) for immunotherapy.

Table 5 shows the prevalence of the individual GLIM criteria. Unintentional weight loss and reduced food intake were the criteria with the highest prevalence, present for approximately one-quarter of the cohort (25.8% and 25.7% respectively).

### 3.3. Inter-Rater Reliability

Inter-rater reliability for malnutrition diagnosis using the GLIM criteria was classified as ‘excellent’ for the overall malnutrition prevalence (GLIM criteria), as well as for each individual criterion (Table 6).

### 3.4. Factors Associated with Malnutrition (Method 1 Scoring)

There was no significant association found between the presence of malnutrition according to the GLIM criteria and sex, age, metastatic disease, social supports, BMI (categorical), treatment setting or treatment location (Table 7). Malnutrition was only found to be significantly associated with BMI as a continuous variable, with participants with lower BMIs more likely to have a malnutrition diagnosis. (Median (IQR) ‘yes’ 24.8 (22.0,27.7) versus ‘no’ 26.3 (23.8, 30.9) *p* = 0.031.)

### 3.5. Prediction of 30 Day Outcomes

A diagnosis of malnutrition using the GLIM criteria was not significantly associated with unplanned hospital admissions (*n* = 21 participants) within the 30 day period from the date of the point prevalence data collection. There was no evidence of multi-collinearity. The full regression model was non-significant, χ*^2^* (8, *n* = 139) = 14.66, *p* = 0.066, indicating that it was not able to predict unplanned hospital admissions. As a whole, the model explained between 10.0% (Cox and Snell R square) and 18.6% (Nagelkerke R square) of the variance in unplanned hospital admissions indicating that other factors not measured in this study contribute to unplanned hospital admissions. The only co-variates which retained significance in the model were age (*p* = 0.034) and the presence of metastatic disease (*p* = 0.011). For each additional year of age participants were 0.95 (0.91, 1.00) times less likely to have an unplanned hospital admission. Participants with metastatic disease were 5.0 (1.6, 17.3) times more likely to have an unplanned hospital admission than participants without metastatic disease.

## 4. Discussion

Using the 2-step approach to diagnosing malnutrition according to the GLIM criteria, this study found that, of the 42% of HNC participants who were at risk of malnutrition, almost one-quarter were malnourished. An excellent level of inter-rater agreement was demonstrated, indicating that the criteria can be reliably applied by experienced clinicians to determine the presence of malnutrition in HNC.

The previously described study using GLIM criteria in HNC by Einarsson et al. reported a malnutrition prevalence of between 0.5 and 32.4% depending on the treatment time point and the combination of GLIM criteria used [8]. Our results fall within this range. However, the secondary analysis approach of our study prevented reporting of the treatment timepoint and required alternate measures to be used for the GLIM criteria. Our use of a surrogate measure for inflammation rather than the objective measure of C-reactive protein used by Einarsson et al. [8] may have resulted in the under-reporting of this criterion, and therefore malnutrition prevalence. Notably, the difference in malnutrition prevalence with and without the use of metastasis to indicate inflammation was minimal. We identified an additional 18 participants who had at least one phenotypic criterion present but did not meet the etiologic criterion. It is possible that these participants may have been classified as malnourished by the GLIM criteria if we had used an objective measure to determine the presence of inflammation, resulting in a higher malnutrition prevalence. Additionally, we subjectively assessed a minimum of four of a possible seven muscle sites to determine a reduction in muscle stores rather than an objective measure such as bioelectrical impedance analysis (BIA) which all may have led to malnutrition under-reporting. However the physical assessment from the PG-SGA has demonstrated the best agreement with the gold standard computed tomography (CT) determined muscularity compared to other surrogate tools used in the clinical setting [30,31], and therefore is a clinically relevant measure. Importantly, the PG-SGA is commonly used in clinical practice in the oncology setting, whilst resources and tools to obtain objective measures, such as CT scans and BIA scales are less available and therefore less likely to be utilised. Further investigation of the validity of the use of the PG-SGA assessment versus objective measures of muscle stores as part of the GLIM criteria is required. 

Our reported GLIM malnutrition prevalence is relatively low compared to previous studies that have used other criteria to diagnose malnutrition in HNC patients. Many of the studies that reported prevalence rates higher than 40% analysed patient data from more than 10 years ago [3,4,5], and therefore the more contemporary patient cohort in our study may account for this difference. Over the last decade the treatment for HNC has improved, including more directed treatments that aim to minimize side effects, as well as improved nutrition practices. The development of evidence-based practice guidelines for the nutritional management of HNC in 2010 [32], along with other oncology focused evidence-based guidelines [33,34], has led to more proactive nutrition interventions including prophylactic insertion of feeding tubes, and improved nutrition outcomes for HNC patients [35,36]. Additionally, our patient cohort comprised almost 80% ambulatory patients and previous studies have shown that ambulatory oncology patients are more likely to have a lower malnutrition prevalence compared to hospitalised inpatients [5,7]. There appeared to be a trend in our sample towards participants from the inpatient treatment setting having higher rates of malnutrition than those in the ambulatory setting (29% versus 18%). Marshall et al. [7] used the PG-SGA to diagnose malnutrition in an Australian oncology cohort in 2012 and 2014 and reported a malnutrition prevalence of 19-25% in ambulatory oncology patients compared to 57% in admitted oncology patients, a finding that is consistent with ours. 

As recommended by the 2-step approach to diagnosing malnutrition using the GLIM criteria, all participants in our study who were not at risk of malnutrition according to the MST were classified as well nourished. This may, however, have led to under-reporting of the malnutrition prevalence. A recent study conducted in a geriatric cohort reported that approximately 15% of participants who were diagnosed as malnourished using the GLIM criteria were not detected by the MST [37]. The MST was reported to be a valid screening tool when compared to the GLIM criteria, but it only had fair sensitivity and specificity. Due to our study being a secondary analysis and the methodology of the original study not requiring a muscle store assessment for those participants not at risk of malnutrition, it was not possible for us to determine a GLIM malnutrition diagnosis for these participants. The MST covers one of the phenotypic (weight loss) and one of the etiologic (poor food intake) criteria of the GLIM criteria. However, the additional components of the GLIM criteria including muscle stores, low BMI and inflammation could lead to a malnutrition diagnosis without scoring the required ≥2 points on the MST. Given this 2-step approach is usual clinical practice, further prospective validation studies are required.

To our knowledge, this is the first study in HNC patients to investigate the inter-rater reliability of applying the GLIM criteria. There was excellent agreement between assessors when applying each of the individual GLIM criterion, as well as the overall GLIM criteria. This finding is important in both the clinical and research settings, as it suggests that using multiple experienced assessors will produce reliable results, if provided with clear, standardised criteria. This can enable health services to compare results across the organisation where multiple clinicians are applying the criteria, as well as within multi-site research studies.

A GLIM malnutrition diagnosis in our study was found to only be significantly associated with BMI. Previous studies conducted in oncology patients using criteria other than GLIM have shown malnutrition to be significantly associated with older age [5,7,16,38], low BMI [7], weight loss >5% [7], hospital admission [5,7], readmission within 30 days of discharge [39], and metastatic disease [5,7]. The relatively small subgroup numbers included in our sample, such as only 30 patients from regional areas, may have contributed to the lack of power to detect an association.

The GLIM Consortium states that for admitted patients, 30 or 60 day readmission are relevant outcomes for use in predictive validity [19]. Again, the relatively small number of readmissions in this cohort may be a contributing factor to the finding that malnutrition diagnosed using GLIM was not associated with readmission in our sample. All participants were alive at 30 day follow up, precluding analysis of the predictive validity of the GLIM criteria for mortality. For survival data, longer-term outcomes may be more relevant to look at in HNC cohorts given in 2012–2016 the five-year relative survival for HNC was 71% [1]. A low prognostic nutritional index has recently been found to be an independent prognostic factor for prediction of one-year mortality in a study of 113 patients with locally advanced HNC [40], suggesting that one year may be a more valid timeframe in HNC cohorts.

This study is only the second study to investigate the prevalence of malnutrition using GLIM criteria, and its predictive validity, in a cohort of HNC patients. The use of data that are commonly collected as part of standard clinical care to apply the GLIM criteria is a strength of this study. However, the use of metastatic disease to indicate inflammation rather than an objective measure is a limitation. This may have led to under-reporting of the presence of this etiologic criterion and therefore malnutrition prevalence. The cut offs used to apply the GLIM criteria differ slightly to those stated in the GLIM criteria, which may also have led to under-reporting. In some participants, anthropometry was self-reported. However, this has been shown to be reliable [41,42]. Additionally, as this was a secondary analysis of a larger cancer malnutrition point prevalence study, additional specific data collection was not possible, including specific treatments and combinations of treatments, longer-term mortality data, and objective data to assess the GLIM criteria. 

## 5. Conclusions

This is the first study to demonstrate excellent inter-rater reliability applying predetermined GLIM criteria to diagnose malnutrition in HNC patients, and one of the first to demonstrate that one in five HNC patients undergoing treatment are malnourished according to the GLIM criteria. Further large, prospective cohort studies are required in HNC patients that use an objective measure to determine inflammation, assess muscle stores for all patients, and apply the GLIM criteria to the whole cohort, as well as to determine the validity of the GLIM criteria to predict patient outcomes, including hospital readmission and one-year mortality.

## Figures and Tables

**Figure 1 nutrients-12-03493-f001:**
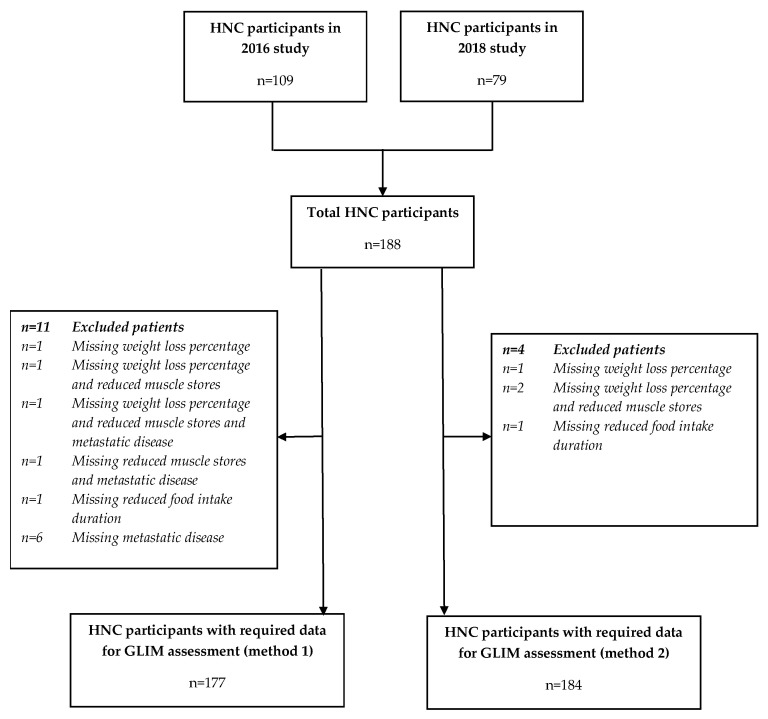
Flowchart of participants.

**Table 1 nutrients-12-03493-t001:** How the GLIM criteria were applied in the present study.

	GLIM Criteria	Present Study
**Phenotypic**	**Weight loss**	
	>5% within past 6 months or >10% beyond 6 months	>5% unintentional weight loss in ≤3 months or ≥4 months
	**BMI**	
	<20 kg/m^2^ if <70 years or <22 kg/m^2^ if >70 years	<20 kg/m^2^ if <70 years or <22 kg/m^2^ if >70 years
	**Reduced muscle mass**	
	Reduced by validated body composition measuring techniques	Evidence of reduced muscle stores at ≥4 muscle sites
**Etiologic**	**Reduced food intake**	
	≤50% of energy requirements for <1 week or any reduction for >2 weeks	<50% of usual intake for >5 days, or <75% of usual intake for >1 month, or >75% of usual intake for >1 month
	**Inflammation**	
	Acute disease/injury or chronic disease-related	Presence of metastatic disease

Abbreviations: GLIM = Global Leadership Initiative on Malnutrition; BMI = body mass index.

**Table 2 nutrients-12-03493-t002:** How the GLIM criteria severity grading was applied in the present study.

	GLIM Criteria	Present Study
**Moderate malnutrition**	**Weight loss**	
	5–10% within past 6 months or 10–20% beyond 6 months	5–10% unintentional weight loss in ≤3 months or ≥4 months
	**BMI**	
	<20 kg/m^2^ if <70 years or <22 kg/m^2^ if >70 years	<20 kg/m^2^ if <70 years or <22 kg/m^2^ if >70 years
	**Reduced muscle mass**	
	Mild-moderate deficit	≥4 muscle sites rated with mild/moderate reduction in muscle stores
**Severe malnutrition**	**Weight loss**	
	>10% within past 6 months or >20% beyond 6 months	>10% unintentional weight loss in ≤3 months or ≥4 months
	**BMI**	
	<18.5 kg/m^2^ if <70 years or <20 kg/m^2^ if >70 years	<18.5 kg/m^2^ if <70 years or <20 kg/m^2^ if >70 years
	**Reduced muscle mass**	
	Severe deficit	≥4 muscle sites rated with severe reduction in muscle stores

Abbreviations: GLIM = Global Leadership Initiative on Malnutrition; BMI = body mass index.

**Table 3 nutrients-12-03493-t003:** Participant characteristics (*n* = 188).

	*n* = 188
**Sex (male)**	139 (73.9)
**Age, years mean (SD)**	63.8 (12.4)
**Social supports**	
Lives alone	37 (19.7)
Lives with family/carer/in residential care	150 (79.8)
Missing	1 (0.5)
**Body mass index, median (IQR)**	25.9 (23.0 – 30.1)
**Presence of metastatic disease (yes)**	25/161 (13.3)
**Current treatment**	
Surgery	80 (42.6)
Chemotherapy	86 (45.7)
Radiotherapy	146 (77.7)
Immunotherapy	3 (1.6)
Other	3 (1.6)
Cancer-related management	4 (2.1)
**Treatment setting**	
Inpatient	41 (21.8)
Ambulatory	147 (78.2)
**Patient status at 30 days**	
Alive Deceased Unknown	169 (89.9) 0 (0) 19 (10.1)

Multiple treatment modalities could be selected for each participant. ‘Other’ included trial drugs.

**Table 4 nutrients-12-03493-t004:** Prevalence of malnutrition risk and malnutrition (*n* = 188).

	Prevalence *n* = 188
**Malnutrition risk (MST ≥ 2)**	
Yes	79 (42.0)
No	109 (58.0)
**Malnutrition prevalence (GLIM criteria)**	
Well nourished	137 (72.9)
Moderate malnutrition	15 (8.0)
Severe malnutrition	25 (13.3)
Missing	11 (5.9)
**Overall malnutrition prevalence (GLIM criteria)**	40/177 (22.6)

A diagnosis of malnutrition was made when at least one phenotypic and one etiologic GLIM criterion were present (method 1). Abbreviations: GLIM = Global Leadership Initiative on Malnutrition; MST = malnutrition screening tool.

**Table 5 nutrients-12-03493-t005:** GLIM criteria prevalence.

	Prevalence
**GLIM phenotypic criteria**	
Unintentional weight loss	47/182 (25.8)
Low BMI	15/188 (8.0)
Reduced muscle stores	41/178 (23.0)
**GLIM etiologic criteria**	
Reduced food intake	48/187 (25.7)
Presence of metastatic disease	13/173 (7.5)

Variations in denominators are reflective of the differing amounts of missing data for each GLIM criterion. Abbreviations: GLIM = Global Leadership Initiative on Malnutrition; BMI = body mass index.

**Table 6 nutrients-12-03493-t006:** Inter-rater reliability.

	*n*	Kappa (95% Lower CI)
**Phenotypic criteria**		
Unintentional loss of weight	181	0.987 (0.962)
Low BMI	187	1.00 (1.000)
Reduced muscle stores	177	0.954 (0.901)
**Etiologic criteria**		
Reduced food intake	186	1.00 (1.000)
Inflammation	171	0.960 (0.882)
**Overall GLIM score**	176	0.985 (0.956)

A diagnosis of malnutrition was made when at least one phenotypic and one etiologic GLIM criterion were present (method 1). Abbreviations: BMI = body mass index; GLIM = Global Leadership Initiative on Malnutrition.

**Table 7 nutrients-12-03493-t007:** Factors associated with malnutrition.

	Malnourished	*p*-Value
**Sex**		1.000
Male	28/139 (20.1)	
Female	10/48 (20.8)	
**Age, years**		1.000
<65	20/100 (20.0)	
≥65	18/87 (20.7)	
**Social supports**		0.377
Lives alone	10/37 (27.0)	
Lives with carer/family/in residential care	28/149 (18.8)	
**BMI, kg/m^2^**		0.216
Underweight	6/18 (33.3)	
Healthy weight	16/71 (22.5)	
Overweight	16/98 (16.3)	
**Presence of metastatic disease**		1.000
Yes	5/25 (20.0)	
No	25/135 (18.5)	
**Treatment setting**		0.164
Inpatient	12/41 (29.3)	
Ambulatory	26/146 (17.8)	
**Treatment location**		0.681
Metropolitan	34/161 (21.1)	
Regional	4/26 (15.4)	

Chi-squared tests were used to assess for significant between-group differences. Values are presented as *n* (%). Variations in denominators are reflective of the differing amounts of missing data associated with each variable. BMI categories: underweight (<18.5 kg/m^2^ if <65 years or <22 kg/m^2^ if ≥65 years); healthy weight (18.5 kg/m^2^ if <65 years or 22–27 kg/m^2^ if ≥65 years); overweight (≥25 kg/m^2^ if <65 years or >27 kg/m^2^ if ≥65 years). Abbreviations: BMI = body mass index.

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
