# Peer review of "Malnutrition Prevalence according to the GLIM Criteria in Head and Neck Cancer Patients Undergoing Cancer Treatment"

_nutrients, 2020, doi:10.3390/nu12113493_

Round 1
Reviewer 1 Report
Review report
The study aims were to determine 1) the prevalence of malnutrition in HNC patients according to the GLIM criteria, using PG-SGA as a subjective assessment of muscle stores; 2) the predictive validity of the GLIM criteria with respect to hospital readmission and mortality; and 3) the inter-rater reliability of using the GLIM criteria to identify malnutrition. Since there are few studies on the new GLIM criteria, the present study add important information, especially on the interrater reliability of using GLIM. However, since the study presents secondary analyses on data from two previously conducted studies, the matching of data to fit the GLIM criteria is poor and the management of data as well as how the results are presented is inconsistent.
Broad comments
Abstract
‘…extracted from a cancer malnutrition point prevalence study was conducted’ should more properly be: two cancer malnutrition point prevalence studies.
For the criterion muscle loss, let the reader know that it was subjective assessment of muscle stores.
‘…and presence of metastatic disease as a proxy for disease burden’ should more properly be: …as a proxy for inflammation.
Introduction
The introduction gives an overview and presentation of head and neck cancer and related nutritional problems. The authors also highlight the problems with no global definition on how to diagnose patients as malnourished.
Consider how the term ‘Malnutrition’ is used in the Introduction, see for example lines 36 and 40. Since no global definition for malnutrition has been available before 2018, the studies cited have used other measures to define patients’ nutritional status, for example, body weight loss percent (ref 2, 3, 5), serum albumin (ref 2, 3), nutritional index (ref 2), anthropometry (ref 3), body mass index (ref 3, 5), and/or Patient-Generated Subjective Global Assessment (ref 4). As the authors address in the Introduction, the diverse use in earlier studies to define patients as malnourished is a problem but it seems like the authors sometimes ‘fall into the same trap’ when using the term malnutrition in the Introduction.
Lines 40-42, describing consequences of malnutrition, a practice guideline from the Clinical Oncology Society of Australia is used as the reference. Consider using the references sited in the guideline instead.
A more proper reference for the sentence in lines 62-63 would be ref 16.
Aim
Aim 2, ‘…determine the predictive validity of the GLIM criteria with respect to hospital readmission and mortality’ could be questioned since the data did not allow for survival analyses (all participant were alive at follow-up). Contrary, results related to factors associated with malnutrition does not show in the aim.
Materials and Methods
Data for the present study derives from previously collected data from two multi-centre malnutrition point prevalence studies that in total included 188 patients and, hence, a satisfactory amount of patients. However, the fact that secondary analyses are performed, the data used does not fit properly to the criteria defined by GLIM. This should be made clearer to the reader in the Materials and Methods and Discussion. This make it even more important that the Material and Methods section is made more distinct when presenting the available data and how the criteria for GLIM have been defined.
It is unclear to the reader where in the treatment trajectory the study sample is i.e. before treatment, during, after, or a combination of all. Also, line 107: what do the authors mean by “none” for treatment approach? Is the statement made in the title and conclusion ‘undergoing treatment’ correct?
No information is given on whether patients with surgery as the sole treatment approach were included/excluded.
Is there any information available on the chemotherapy, radiotherapy, or immunotherapy used?
The GLIM criteria:
- Another cut-off for timeframe than suggested by GLIM was used in the present study i.e. three months (present study) compared to six months (GLIM). This should be made more clearer to the reader, both in text and Figure 1.
- Cut-offs used for BMI should be stated: both the cut-offs for GLIM and cut-offs for underweight, normal weight, and overweight (categorical variable used in the regression model).
- It is unclear why not all the muscle sites specified in the PG-SGA tool were assessed (address in the Material and Methods) and how this could have impacted the result (address in the Discussion).
- How was the etiologic criteria ‘Reduced food intake’ defined in the present study. This should be made clearer in the text and Figure 1.
Statistics: Considering using quartiles instead of inter-quartile range as measures of spread. Please clarify if the co-variate ‘age’ was used as a categorical or continuous variable in the regression model.
Results
It is an inconsistent use of number of participants. Of the 188 HNC participants in original datasets, 177 participants had the required data to be able to determine malnutrition according to GLIM (outcome variable). Hence, the study cohort consists of n = 177 and this number should be used throughout the presentation.
- Different numbers of patients in the different tables makes it confusing for the reader:
- Table 1 & 2: present participant characteristics and prevalence of malnutrition risk and malnutrition for n = 177 instead of n = 188.
- Table 3 & 5 could be made more distinct if choosing n = 177.
- Lines 190-191: ‘The overall prevalence of malnutrition according to the GLIM criteria was 22.6%’. Lines 194-196 ‘The overall malnutrition prevalence according to the GLIM criteria, omitting the use of metastatic disease as a proxy criterion for inflammation and using only the presence of reduced food intake as the etiologic criterion was 20.4% (38/187)’. The first prevalence of malnutrition is counted from n = 177 and the latter from n = 187 which make the two numbers incomparable.
- Lines 199-201: since the patients characteristics are presented from n = 188 instead of n = 177, the numbers for each treatment modality does not compare with the numbers found in Table 1.
Please clarify to the reader what it means with ‘primary treatment recorded’ (line 199).
Section 3.3: please clarify to the reader what malnutrition prevalence that is used i.e. overall malnutrition prevalence or prevalence using only the presence of reduced food intake as the etiologic criterion. It is unclear why lines 198-201 are not added to this section instead (and Table 4) with statistical significant associations between treatment groups.
How many patients had unplanned hospital admissions within the 30-day period from the date of the point prevalence data collection?
As stated earlier, lines 226-227 can be highly questioned.
Table 1 could be made more distinct and clear to the reader i.e.:
- Spell out abbreviations.
- Let the reader know what current treatment ‘other’ and ‘none’ means.
- Think about where the n (%), mean (SD), and median (IQR) should be placed. It is confusing when n (%) is used as the header for the column to the right when not all numbers in that column shows n (%).
Some information needs to be added in Table 2 to make it possible to interpret the table independent from the text i.e. more information in the table heading, how malnutrition risk was assessed, GLIM criteria used, and related cut-offs. Some information could preferable be added in a footnote below the table.
Some information needs to be added in Table 3 to make it possible to interpret the table independent from the text i.e. more information in the table heading, cut-offs used for each GLIM criteria, and abbreviations should be spelled out. Some information could preferable be added in a footnote below the table.
Table 4 could be made more distinct and clear to the reader i.e.:
- Spell out abbreviations.
- Check the heading and add information on how malnutrition was assessed so the table can be interpreted independently without the text.
- Why was 65 years chosen as the cut-off for age?
- Check grammar in the footnote.
Some information needs to be added in Table 5 to make it possible to interpret the table independent from the text i.e. more information in the table heading, cut-offs used for each GLIM criteria, and abbreviations should be spelled out. Some information could preferable be added in a footnote below the table.
Discussion
The Discussion contains a satisfactory presentation of how the results of the present study can be interpreted in the light of previous research. The authors address the weakness of secondary analyses and how this could have impacted the result. It is important that all GLIM criteria that have not been properly used (weight loss, food intake?) are addressed when discussing study limitations.
As stated earlier in the manuscript, multiple treatment modalities could be selected for each participant. Many patients treated for head and neck cancer receive a multimodal treatment approach and this was not possible to study using the previously collected data.
Study sample should be addressed more properly. Also, the fact that some of the anthropometry values were self-assessed by the patients should be acknowledged.
Reviewer 2 Report
The authors did an excellent job explaining the GLIM criteria and their study. I agree that a lack of power likely impeded their ability to find significant association between patient outcomes and malnutrition. My only suggestion would be to shift section on inter-rater reliability ahead of the sections on factors associated with malnutrition and prediction of 30-day outcomes. I think this would give better flow and allow you to highlight the positive result earlier in the paper.
Reviewer 3 Report
In the paper Malnutrition prevalence according to the GLIM criteria in head and neck cancer patients undergoing cancer treatment, the problem of the variation in malnutrition prevalence of NHC patients and that numerous criteria are used to diagnose malnutrition is presented. A secondary analysis study using previously collected data from a multi-site cancer malnutrition prevalence audit of hospitalised and ambulatory patients.
Strengths:
The aims are well defined and significant.
An interesting and relevant study.
The authors have specified how the GLIM criteria indicators were measured and offer justification and took into account when applying proxy measures and subjective over objective measures with regard to methodology and reporting and interpreting of outcomes.
The results offer an advance on current knowledge in that the authors seek to investigate the ability of GLIM criteria to predict clinically relevant outcomes in this population of NHC patients. Also, to report on inter-rater reliability on categorisation of malnutrition using GLIM criteria, both of which there is no current literature.
Weaknesses:
- Whilst the authors have specified how variables and GLIM criteria applied, there is some confusion when reading the methodology and results.
2.3.2 malnutrition screening 2.3.3 Anthro: it is specified that patients with MST score 2 or greater where at risk of malnutrition and therefore progressed to nutrition assessment and subjective assessment of muscle stores. So, the reader would assume only patients who had MST 2+ would have been included in dataset for diagnosis of malnutrition according to GLIM. The authors make note of this in their discussion (line 277 -278).
However, results table 2 (line 192) malnutrition risk prevalence is n=79. It was not clear to the reader how 79 patients had positive MST, but 188 patients had GLIM criteria applied.
There could also be clearer description around time frame of weight loss and time frame of reduced food intake used in the application of the GLIM criteria in figure 1. What time frame cut offs of weight loss was used for >5% weight loss, 5-10%, >10% weight loss in figure 1?
What time frame cut offs were used for reduced food intake in figure 1?
- Consider reporting on malnutrition prevalence according to PG-SGA in this same study group. The methodology reads that PG-SGA and a malnutrition diagnosis was recorded during the original malnutrition point prevalence studies? Would PG-SGA be considered a semi gold standard to compare against GLIM criteria?
- The grouping of hospitalised and ambulatory/community patients together. Outcomes measures 30-day readmission and mortality are reported. This would be recommended if the population was mostly hospitalised, however hospitalised patients only make up 22% of the study participants. One-year readmission and mortality data would be more appropriate for ambulatory patients. The original dataset was 2016 and 2018, is one -year mortality available? Consider adding this as a limitation, 30-day mortality rates do not make sense for ambulatory outpatients.
- Line 153-154: unplanned hospital admission up to 30 days post GLIM scoring. Clarify does this mean the original data collection date 2016/2018, and not the 30 days post the study investigators retrospectively scoring the GLIM criteria?
- Line 223. For each additional year of age participants were 0.95 (0.91, 1.00) times less likely to have an unplanned hospital admission (p = 0.034).
Is this correct? That for every year older, the older patient has a 5% less chance of unplanned hospital admission.
- Table 1. Line 185, format to ensure the table is displayed in full on one page, rather than over 2 pages as makes it difficult to read.
Round 2
Reviewer 1 Report
The authors have made satisfactory changes to the manuscript which have made the presentation of data more consistent. The authors have made the results more clearer using 'method 1' and 'method 2' for diagnosis of malnutrition when at least one phenotypic and one etiologic criterion were present or when excluding the presence of metastatic disease (proxy inflammation criterion), respectively. They have also added two tables to further describe how the data from the present study were used to assess malnutrition according to GLIM. The main limitations of the study is the missing data for various variables and the secondary data analysis for the diagnosis of malnutrition using GLIM.
In the Introduction, lines 38-39 and 42. Considering using either ‘various’ or ‘different’.
In the Materials and Methods, lines 126-128, add a reference for the cut-offs used for BMI.
In Table 1, reduced food intake. Which ones were used to define presence of the etiologic criterion? Only that cut-off should be presented in the table.
Tables 4 & 5, footnote. ‘Values are presented as n (%) unless otherwise stated.’ can be removed.
